# Genome sequencing unveils a regulatory landscape of platelet reactivity

Ali R. Keramati [1,2,124], Ming-Huei Chen[3,4,124], Benjamin A. T. Rodriguez[3,4,5,124], Lisa R. Yanek [2,6], Arunoday Bhan[7], Brady J. Gaynor [8,9], Kathleen Ryan[8,9], Jennifer A. Brody [10], Xue Zhong[11], Qiang Wei[12], NHLBI Trans-Omics for Precision (TOPMed) Consortium*, Kai Kammers[13], Kanika Kanchan[14], Kruthika Iyer[14], Madeline H. Kowalski[15], Achilleas N. Pitsillides[4,16], L. Adrienne Cupples [4,16], Bingshan Li [12], Thorsten M. Schlaeger[7], Alan R. Shuldiner[9], Jeffrey R. O'Connell[8,9], Ingo Ruczinski[17], Braxton D. Mitchell [8,9], Nauder Faraday[2,18], Margaret A. Taub[17], Lewis C. Becker[1,2], Joshua P. Lewis [8,9,125✉], Rasika A. Mathias [2,14,125✉] & Andrew D. Johnson [3,4,125✉]

Platelet aggregation at the site of atherosclerotic vascular injury is the underlying pathophysiology of myocardial infarction and stroke. To build upon prior GWAS, here we report on 16 loci identified through a whole genome sequencing (WGS) approach in 3,855 NHLBI Trans-Omics for Precision Medicine (TOPMed) participants deeply phenotyped for platelet aggregation. We identify the *RGS18* locus, which encodes a myeloerythroid lineage-specific regulator of G-protein signaling that co-localizes with expression quantitative trait loci (eQTL) signatures for *RGS18* expression in platelets. Gene-based approaches implicate the *SVEP1* gene, a known contributor of coronary artery disease risk. Sentinel variants at *RGS18* and *PEAR1* are associated with thrombosis risk and increased gastrointestinal bleeding risk, respectively. Our WGS findings add to previously identified GWAS loci, provide insights regarding the mechanism(s) by which genetics may influence cardiovascular disease risk, and underscore the importance of rare variant and regulatory approaches to identifying loci contributing to complex phenotypes.

[1] Division of Cardiology, Johns Hopkins University School of Medicine, Baltimore, MD, USA. [2] GeneSTAR Research Program, Johns Hopkins University School of Medicine, Baltimore, MD, USA. [3] Division of Intramural Research, Population Sciences Branch, National Heart, Lung and Blood Institute, Bethesda, MD, USA. [4] The Framingham Heart Study, Framingham, MA, USA. [5] Valo Health, Boston, MA, USA. [6] Division of General Internal Medicine, Johns Hopkins University School of Medicine, Baltimore, MD, USA. [7] Boston Children's Hospital, Boston, MA, USA. [8] Division of Endocrinology, Diabetes, and Nutrition, University of Maryland School of Medicine, Baltimore, MD, USA. [9] Program in Personalized and Genomic Medicine, University of Maryland School of Medicine, Baltimore, MD, USA. [10] Cardiovascular Health Research Unit, University of Washington School of Medicine, Seattle, WA, USA. [11] Vanderbilt Genetics Institute, Division of Genetic Medicine, Department of Medicine, Vanderbilt University Medical Center, Nashville, TN, USA. [12] Vanderbilt Genetics Institute, Department of Molecular Physiology and Biophysics, Vanderbilt University, Nashville, TN, USA. [13] Biostatistics and Bioinformatics, Oncology, Sidney Kimmel Comprehensive Cancer Center, Johns Hopkins University School of Medicine, Baltimore, MD, USA. [14] Division of Allergy and Clinical Immunology, Johns Hopkins University School of Medicine, Baltimore, MD, USA. [15] Department of Biostatistics, University of North Carolina, Chapel Hill, NC, USA. [16] Department of Biostatistics, School of Public Health, Boston University, Boston, MA, USA. [17] Bloomberg School of Public Health, Biostatistics, Johns Hopkins University, Baltimore, MD, USA. [18] Department of Anesthesiology and Critical Care Medicine, Johns Hopkins University School of Medicine, Baltimore, MD, USA. [124]These authors contributed equally: Ali R. Keramati, Ming-Huei Chen, Benjamin A.T. Rodriguez. [125]These authors jointly supervised this work: Joshua P. Lewis, Rasika A. Mathias, Andrew D. Johnson. *A list of authors and their affiliations appears at the end of the paper. ✉email: jlewis2@som.umaryland.edu; rmathias@jhmi.edu; johnsonad2@nhlbi.nih.gov

A therosclerotic cardiovascular diseases (ASCVD) have remained the major cause of morbidity and mortality worldwide. The hallmark of ASCVD is aggregation of activated platelets on a ruptured atherosclerotic plaque followed by thrombus formation[1]. Hemostasis and platelet aggregation is an evolutionary conserved process that is maintained by a delicate balance between agonists like ADP and epinephrine and antagonists like prostaglandins[2]. Prior studies have shown that platelet aggregation in response to agonists is highly heritable with heritability estimates between 40 and 60%[3–5]. High platelet reactivity at baseline and after inhibition with aspirin is associated with poor cardiovascular outcome[6,7]. Antiplatelet therapies are standard-of-care for secondary prevention of the complications of occlusions in coronary, cerebral, and peripheral arteries. Prior genome- and exome-wide association studies have identified at least 8 common variants for platelet aggregation in response to different agonists[8–11]. With the exception of a few limited gene-based scans[9,12], no previous genome-wide studies have systematically evaluated the contribution of both common and rare variants to heritability of agonist-induced platelet reactivity. Thus, it is likely that significant missing heritability remains for platelet function traits.

In this work leveraging the scientific resources of the NHLBI Trans-Omics for Precision Medicine (TOPMed) Program, we report the first association study of platelet aggregation in response to variety of physiological stimuli using whole-genome sequencing (WGS) data. We sought to 1) refine previously identified GWAS loci, 2) identify novel loci that determine platelet aggregation in response to different doses of ADP, epinephrine and collagen, 3) examine the collective burden of coding variants on platelet aggregation, and 4) evaluate the collective burden of rare non-coding variants of megakaryocyte-specific super-enhancer regions on platelet aggregation. Extension of genetic findings using biobank resources as well as ex vivo cell-based functional systems were also performed.

## Results

**Single-variant based tests for association.** There were a total of 19 harmonized phenotypic measures of platelet aggregation evaluated in this investigation (Supplementary Table 1). This includes 9 phenotypes for adenosine diphosphate (ADP) as an agonist, 9 for epinephrine, and 4 for collagen. Genome-wide single variant tests for association were performed on ~28 million variants in 3,125 European Americans (EA) and 730 African Americans (AA) (Supplementary Table 2) from the Framingham Heart Study (FHS), Older Order Amish Study (OOA), and the Genetic Study of Atherosclerosis Risk (GeneSTAR). We identified 101 variants associated with platelet aggregation in response to ADP, epinephrine, or collagen ($P$ value $< 5 \times 10^{-8}$, Fig. 1, Supplementary Fig. 1). Using iterative conditional analyses, genome-wide significant variants were refined down to 16 independent loci (Table 1). With the exception of two variants (rs12041331 and chr17:21960955) all loci were associated with platelet aggregation in response to a single agonist (Fig. 1B), and most of the identified loci were not present in the prior array-based approaches[8–10] (Table 1, Supplementary Figs. 2–4).

**Replication of the single-variant results.** Replication of discovery findings was performed in up to 2,009 independent samples from FHS, OOA, and GeneSTAR (Supplementary Data 1), and extended into an independent cohort (the Caerphilly Prospective Study [CaPS], $N = 1183$) for ADP and collagen-induced platelet aggregation phenotypes[8,13] (Supplementary Table 4). Among the 7 previously reported loci[10], 2 were replicated in this investigation (PEAR1 and ADRA2A, Table 1).

Reduction in sample size, a low overlapping percentage (<75%) of participants in 2 of the previously studied cohorts (FHS and GeneSTAR European samples), addition of subjects (OOA and GeneSTAR African Americans), and the difference between WGS data and HapMap imputed dosage data (Supplementary Table 5a) may explain, in part, the lack of association observed with the other 5 previously identified loci. Meta-analysis, as opposed to mega-analysis approaches, did not meaningfully change the interpretation of these findings (Supplementary Table 5a, b) comparing the current WGS results to prior studies. Comparison of previous results with the current investigation for the RGS18 variant is shown in Supplementary Table 5b; all other newly-identified WGS variants from this study were not available in the previous investigation.

**Co-localization of the genetic loci with eQTLs in platelets.** Given that all 16 loci identified using single-variant approaches are located in non-coding regions of the genome, we tested for co-localization between these regions and eQTL data available through RNA sequencing of platelets in 180 European Americans from GeneSTAR (Supplementary Table 6). We found that sentinel variants in the PEAR1 and RGS18 loci were eQTLs for PEAR1 and RGS18, respectively. No co-localization was noted for any of the remaining 14 loci. As noted in Fig. 2a, there is likely only a single variant accounting for the PEAR1 GWAS peak, in contrast to RGS18 where there are likely several causal variants.

**PheWAS in external Biobanks.** An examination of the sentinel variants reported in Table 1 was performed in the UK Biobank and BioVU as presented in Supplementary Data 2. The minor allele (A) of PEAR1 at rs12041331, which is known to be associated with lesser platelet aggregation, PEAR1 RNA, and protein expression[14,15], was associated with increased odds of gastrointestinal bleeding in both EAs and AAs in the BioVU Biobank PheWAS.

**Functional follow up of the RGS18 locus.** In the RGS18 region, several variants were replicated using independent samples (Supplementary Data 1), and additional evidence was also observed for ADP and collagen aggregation phenotypes in the CaPS study (Supplementary Table 4). Overlaying the associated variants with platelet eQTLs and megakaryocytic epigenome features, there are several potential candidate polymorphisms (Supplementary Table 8). Consistent with our human results, independent $Rgs18^{-/-}$ mouse studies suggest Rgs18 inhibits pre-agonist stimulated platelet reactivity, with knockouts exhibiting exaggerated platelet reactivity to multiple agonist pathways, decreased bleeding times, and increased arterial occlusion[16,17]. This is attributed to a loss of inhibition of multiple G-protein coupled receptor signaling pathways in platelets[18]. The minor allele (C) of RGS18 at rs1175170, is associated with arterial thrombosis/embolization in both EAs and AAs in the BioVU BioBank (Supplementary Data 2). Allele-specific and transcription-factor overexpression studies suggest that rs12070423, which may disrupt a GATA1 target site, and rs4495675, which may disrupt a NFE2 target site, both reduce RGS18 expression (Fig. 2b, Supplementary Table 9, Supplementary Figs. 4 and 5). These SNPs are in LD (minimum $r^2$ 0.614) with rs1175170 suggesting they may be functional variants on the same haplotype that affect RGS18-mediated platelet activation.

**Genes identified through rare variant based approaches.** SKAT[19] gene-based tests using a MAF threshold of 0.05 were

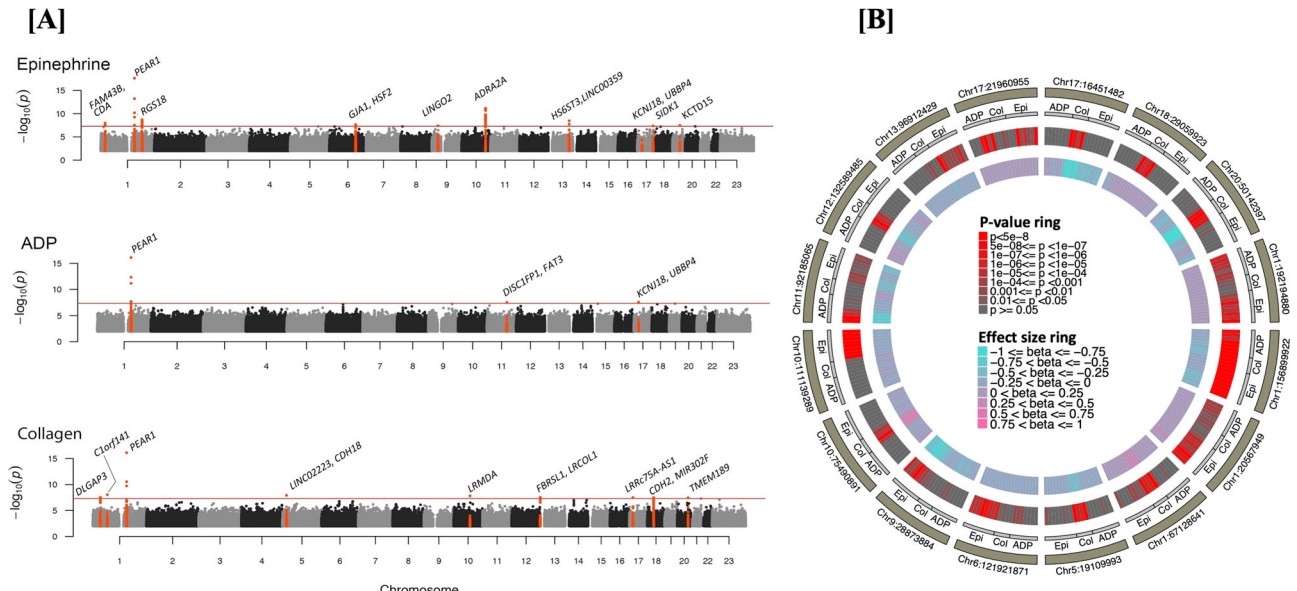

**Fig. 1 Genome-wide association study results for platelet aggregation and summary effects at 16 loci with $P < 5 \times 10^{-8}$. A** Genome-wide association study results for platelet aggregation in response to epinephrine, ADP and collagen in 3855 TOPMed participants. *P* values presented are a summary across all individual phenotypes for the single agonist (i.e., the minimum *P* value for the variants from 8, 7 and 4 individual phenotypes for epinephrine, ADP, and collagen, respectively described in Supplementary Table 1). *P* values are from a two-sided score test with no adjustment for multiple testing in panels **A** and **B**. Loci passing genome-wide significance ($P < 5 \times 10^{-8}$) are marked by red dots. Locus names represent the nearest (for novel) or previously annotated (for known) gene. The red line indicates a *P* value threshold of $5 \times 10^{-8}$, corresponding to genome-wide significance. **B** Circle plot of the sentinel variant at the 16 loci for each of the 19 phenotypes showing strength of GWAS signal (second from center ring) and magnitude/direction of effect (center ring).

conducted for deleterious variants mapping to 17,774 protein-coding genes (Supplementary Table 10, Supplementary Fig. 6) with significant findings after Bonferroni correction for *SVEP1* (ADP-induced platelet aggregation, *P* value = $2.6 \times 10^{-6}$), *BCO1* (epinephrine-induced platelet aggregation, $P = 8.9 \times 10^{-7}$), *NELFA* (collagen-induced platelet aggregation, $P = 1.7 \times 10^{-6}$) and *IDH3A* (collagen-induced platelet aggregation, *P* value = $2.6 \times 10^{-6}$). Through *leave-one-out* analysis, we observed that these associations were driven mainly by single or limited sets of rare variants (Supplementary Fig. 7, Supplementary Table 11). For example, the *SVEP1* association with ADP-induced platelet aggregation was solely driven by a nonsynonymous variant (Gly229Arg) in the second exon (rs61751937, MAF 0.028, *P* value = $5.8 \times 10^{-6}$). This variant alters a highly conserved residue located in the protein's VWFa domain (Fig. 3A–C). The finding remained significant in the replication cohort (*P* value = 0.004, Supplementary Table 3) and CaPS (*P* value = 0.008, Supplementary Table 4), both of which demonstrated an association with increased ADP-induced platelet reactivity. Both variants are modestly associated with CVD outcomes in the UK BioBank (Supplementary Table 7).

**The role of genetic variants in MK-specific super-enhancers**. To investigate the role of genetic variation on regulatory importance in the context of super-enhancers, we aggregated rare non-coding variants across a set of 1,065 published MK-specific super-enhancers (Supplementary Fig. 8)[20]. We found rare non-coding variants in a super-enhancer at the *PEAR1* locus were significantly associated with ADP- ($P = 2.4 \times 10^{-8}$), epinephrine- (*P* value = $1.1 \times 10^{-7}$) and collagen- (*P* value = $2.7 \times 10^{-5}$) induced platelet aggregation. We observed, in marked contrast to our gene-based coding variant analyses, that the association signal in the *PEAR1* super-enhancer is driven by multiple rare variants in the region (Supplementary Fig. 9).

## Discussion

In this WGS study of platelet aggregation, we identify and replicate several loci contributing to trait variation. A WGS approach continues to validate the importance of the *PEAR1* locus. Previous work demonstrated a single, common (~14% MAF) intronic peak variant in *PEAR1* (rs12041331) is associated with platelet phenotypes using GWAS, regional sequencing, and exonic approaches[8,12,14,21]. The minor allele of rs12041331 is linked to decreased PEAR1 platelet protein levels[14,15], potentially through alteration of a methylation site in MKs[22]. In addition, the role of this gene in platelet signaling is supported by mechanistic studies[23,24]. Here, a sequencing-based approach followed by co-localization with platelet eQTLs reveal that results are consistent with a model that a single, common causal variant explains the platelet reactivity signal with respect to *PEAR1*. Similar to the case of *PEAR1*, we recently identified a single strong regulatory SNP, rs10886430 intronic to *GRK5*, that affects a GATA1 transcription factor site and regulates platelet gene expression in a highly cell-type specific manner, ultimately accounting for ~20% of variation in thrombin-platelet reactivity via PAR4 receptor regulation, and being causally related to both venous and arterial disease risk[11]. These examples demonstrate how single SNPs of large effect can be identified and ultimately associated with CVD endpoints but require detailed studies of agonist-specific phenotypes and cell-specific expression patterns that will otherwise be missed.

The proteins RGS10 and RGS18 are highly expressed in platelets and are important regulators of G protein signaling that plays a role in multiple pathways of activation in platelets. Our results indicate common *RGS18* platelet regulatory alleles modulate human platelet function likely through GATA1 and/or NFE2 interacting sites. Furthermore, our findings in independent biobanks and ancestry groups that the allele that leads to increased platelet reactivity is also associated with cardiovascular

**Table 1 Sixteen loci identified through single variant approaches for genome-wide association for platelet aggregation in response to epinephrine, ADP and collagen in 3855 TOPMed participants.**

| Known vs Novel[a] | chr:pos (hg38) | rsID | ref/alt | Nearest Gene | MAF | ADP | | Collagen lag time | | Epinephrine | |
|---|---|---|---|---|---|---|---|---|---|---|---|
| | | | | | | P | beta | P | beta | P | beta |
| Novel | 1:20567949 | rs2137738 | A/T | FAM43B,CDA | 0.077 | 2.62E−04 | 0.22 | 1.71E−02 | −0.108 | 1.04E−08 | 0.306 |
| Novel | 1:67128641 | rs142001088 | C/T | C1orf141 | 0.018 | 2.88E−02 | 0.203 | 9.25E−09 | −0.503 | 2.51E−03 | 0.276 |
| Known | 1:156899922 | rs12041331 | G/A | PEAR1 | 0.148 | 7.61E−17 | −0.329 | 7.58E−17 | 0.317 | 2.31E−18 | −0.358 |
| Novel | 1:192194880 | rs1175170 | G/C | RGS18,RGS21 | 0.442 | 7.86E−06 | 0.123 | 2.37E−02 | −0.057 | 1.96E−09 | 0.155 |
| Novel | 5:19109993 | rs112157462 | T/C | LINC02223, CDH18 | 0.022 | 1.64E−02 | −0.281 | 1.19E−08 | 0.458 | 5.90E−03 | −0.23 |
| Novel | 6:121921871 | rs58250884 | A/G | GJA1,HSF2 | 0.087 | 1.69E−03 | −0.153 | 2.34E−02 | 0.106 | 2.22E−08 | −0.273 |
| Novel | 9:28873884 | rs185159562 | T/A | LINGO2 | 0.005 | 1.16E−02 | −0.447 | 1.45E−01 | 0.243 | 3.87E−08 | −0.988 |
| Novel | 10:75490891 | rs138028657 | A/G | LRMDA | 0.006 | 6.44E−01 | 0.073 | 1.52E−08 | −0.858 | 6.14E−01 | −0.102 |
| Known | 10:111139289 | rs7097060 | T/A | ADRA2A,GPAM | 0.137 | 5.45E−01 | −0.028 | 6.73E−01 | −0.015 | 6.68E−12 | −0.251 |
| Novel | 11:92185065 | rs183146849 | A/T | DISC1FP1,FAT3 | 0.012 | 3.11E−08 | −0.702 | 4.38E−01 | −0.084 | 5.73E−04 | −0.376 |
| Novel | 12:132589485 | rs140148392 | G/A | FBRSL1,LRCOL1 | 0.009 | 1.51E−01 | 0.254 | 2.88E−08 | 0.669 | 3.54E−01 | 0.138 |
| Novel | 13:96912429 | rs61974290 | A/G | HS6ST3, LINC00359 | 0.057 | 9.58E−02 | −0.088 | 9.43E−03 | 0.134 | 3.36E−09 | −0.4 |
| Novel | 17:16451482 | rs575524466 | G/A | LRRC75A-AS1 | 0.003 | 1.90E−01 | −0.315 | 3.13E−08 | 1.169 | 5.35E−02 | −0.463 |
| Novel | 17:21960955 | − | A/T | KCNJ18,UBBP4 | 0.276 | 2.87E−08 | 0.167 | 1.88E−03 | −0.089 | 7.73E−19 | 0.26 |
| Novel | 18:29059923 | rs138845468 | TAAATA/T | CDH2,MIR302F | 0.082 | 6.99E−02 | 0.087 | 3.23E−08 | −0.25 | 1.21E−01 | 0.087 |
| Novel | 20:50142397 | rs542707094 | CTG/C | TMEM189, TMEM189-UBE2V1 | 0.003 | 3.16E−02 | −0.486 | 3.53E−08 | 1.194 | 3.05E−02 | −0.487 |

P values presented are a summary across all individual phenotypes for the single agonist (i.e., the minimum P value for the SNV from 8, 7 and 4 individual phenotypes for epinephrine, ADP, and collagen, respectively), alternate allele (alt) represents minor allele (see Supplementary Table 1). P values are from a two-sided score test with no adjustment for multiple testing.
[a]Loci were defined as known if they were identified in the prior array-based GWAS approaches[8–10], else they were labeled as novel.

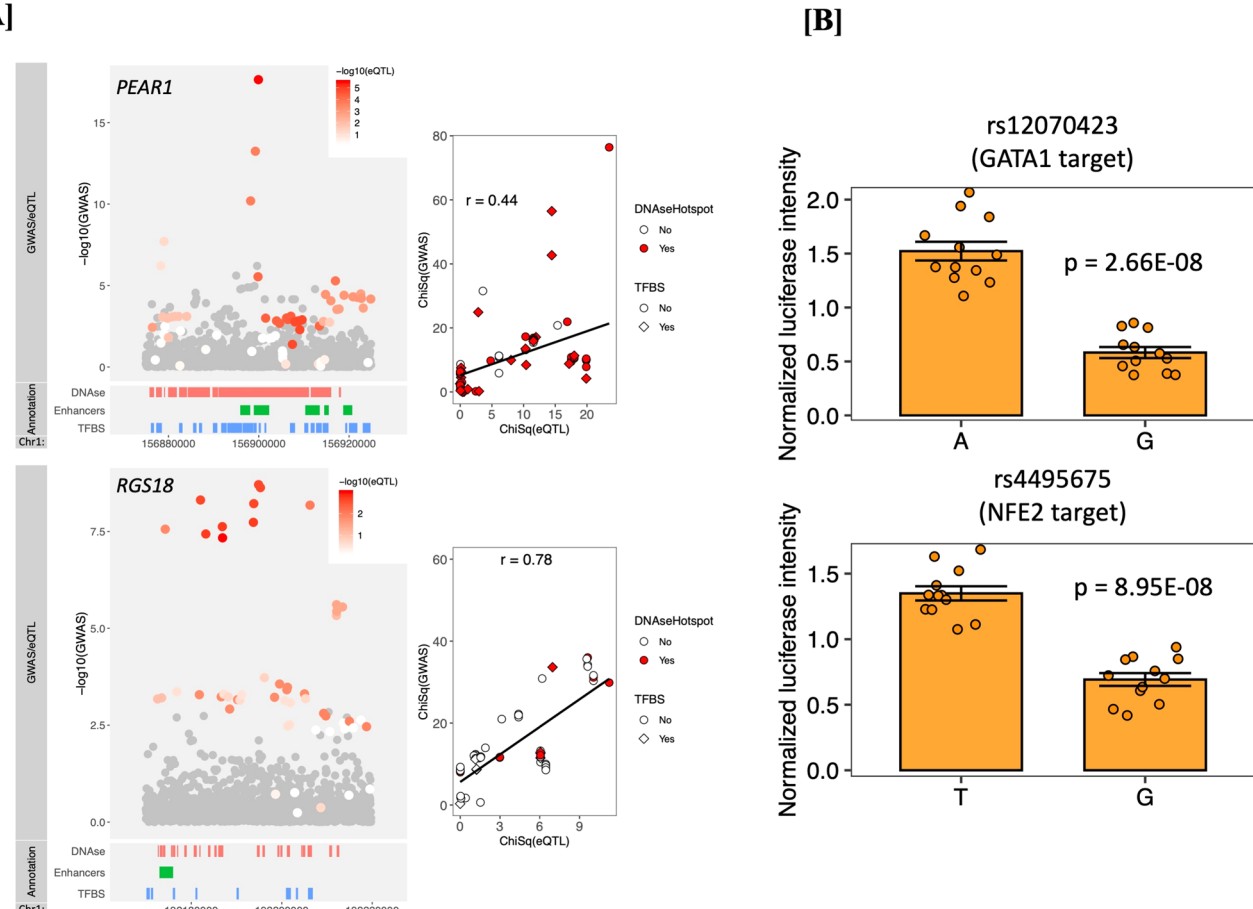

**Fig. 2 Co-localization of WGS signals in *PEAR1* and *RGS18* regions with effects on platelet gene expression and regulatory features and demonstration of *RGS18* allele-specific SNP effects on enhancer activity.** Co-localization of WGS association signal and platelet eQTL signatures and allele-specific experiments for *RGS18* SNP enhancers. *P* values in panel **A** are two-sided score/linear model tests for GWAS/eQTLs, respectively, with no adjustment for multiple testing. *P* values in Panel **B** are from a two-sided Welch test with no adjustment for multiple testing. **A** Top panels show co-localization between *PEAR1* eQTL and WGS association for platelet aggregation in response to Epi_low1 (see Supplementary Table 1) and bottom panels are between *RGS18* eQTL with WGS association for platelet aggregation in response to Epi_low 5 (see Supplementary Table 1). In the left panels, the region of co-localization is zoomed to the sentinel SNV ±25 kb, the Y axis shows the -log(P) of the GWAS association, the color of the dot represents the strength of the eQTL evidence for the gene, and SNVs that were not included in eQTL analysis are shown in gray. The right panels show the scatter plot and correlation between the ChiSquare statistic for the GWAS and eQTL signal for all SNVs present in both sets of data. **B** Allele-specific enhancer activity differences for rs12070423 A or G alleles in HEK293 cells lentiviral transfected to overexpress GATA1 (top) and rs4495675 T or G alleles in HEK293 cells lentiviral transfected to overexpress NFE2 (bottom). Each allele-specific result represents results of 12 experiments (12 biological replicates over 3 independent replicates). Data presented represent mean values ± SEM.

and thrombotic outcomes including occlusions, cerebrovascular disease, cardiac arrest, embolism and deep vein thrombosis suggest that RGS18 may be a critical node for intervention in platelets.

The WGS approach allowing for a rare-variant gene-based analysis suggests that *SVEP1* may have previously unappreciated and multifactorial roles in contributing to CVD. Homozygous *Svep1*$^{-/-}$ mice die from edema, and heterozygous mice, as well as zebrafish, experience arterial and lymphatic vessel malformations[25–27]. Consistent with previous investigations[28], RNAseq data in a subset of GeneSTAR participants do not indicate expression of *SVEP1* in platelets. We find that rs61751937 is the strongest plasma protein QTL for SVEP1 (*P* value = $5.2 \times 10^{-64}$), reducing expression[29], suggesting the effects may be mediated through interactions of platelets with other cell types in circulation. This conserved protein could potentially affect platelet function and CVD through several mechanisms including cell-cell adhesion, cell differentiation, and functions in bone marrow niches[30]. Recent functional work demonstrates that

SVEP1 is expressed in plaques. Further experiments suggested that deficiency of Svep1 affects Cxcl1 endothelial release and promotes proinflammatory leukocyte recruitment to plaques[31]. Given our assays are ex vivo assessments of platelet function in PRP lacking endothelial, leukocyte and smooth muscle cells, this suggests that alteration of SVEP1 levels or other related factors in plasma may also have direct effects on platelets that may influence thrombus formation.

In conclusion, there is a large body of evidence supporting the hypothesis that hyper-reactive platelets may predict future thromboses in both healthy individuals[6] and those who have already experienced thrombosis[32,33]. Therefore, better understanding of the genetic determinants of heightened platelet aggregation is likely critical in the early prediction of thrombosis events as well as aiding in pharmacogenetic efforts pertaining to antiplatelet therapy. By applying contemporary WGS strategies in participants with extensive platelet reactivity phenotype data, we show the potential for such approaches to identify genetic determinants that may impact such traits.

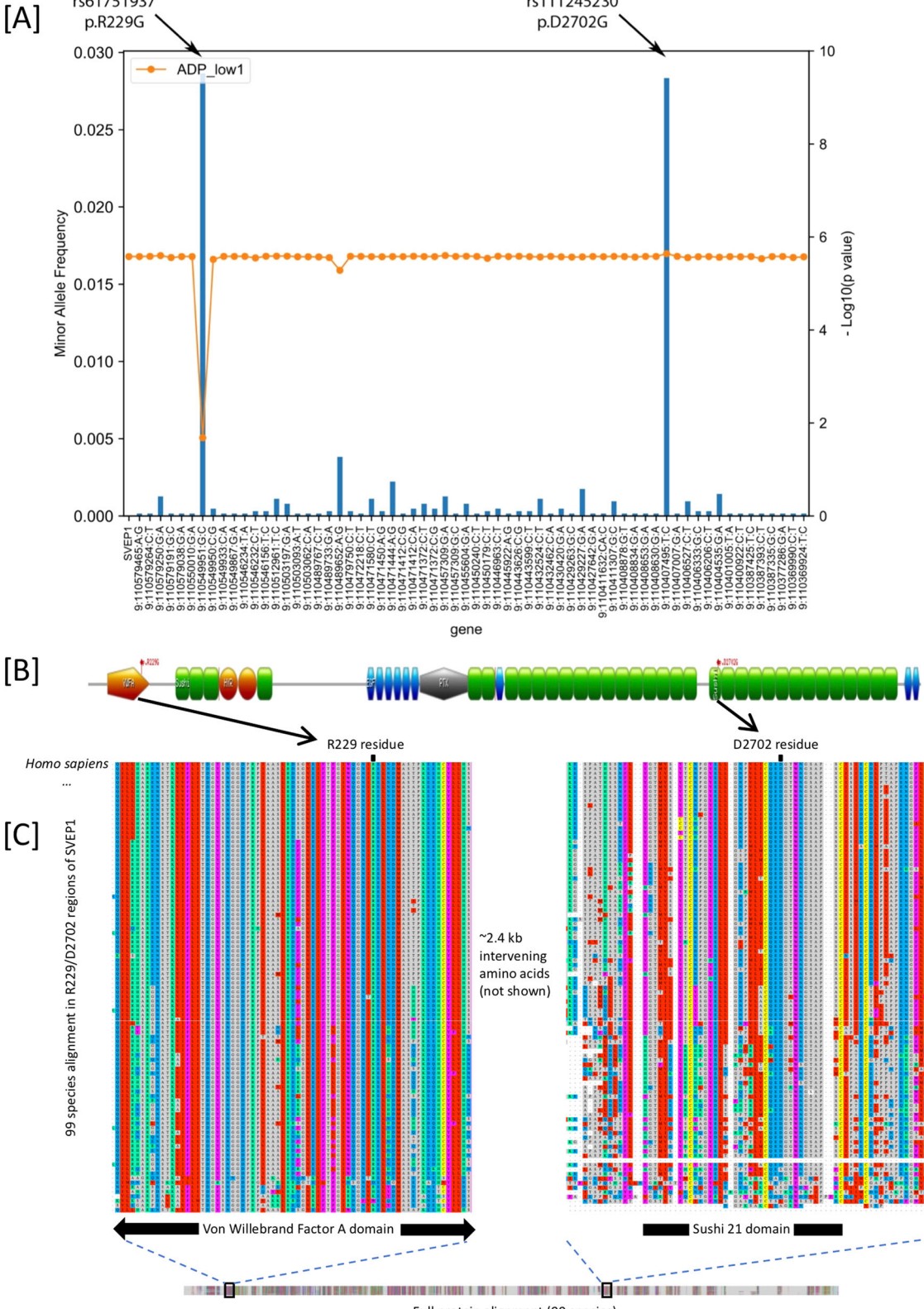

Full protein alignment (99 species)

## Methods

### Description of study populations

*GeneSTAR.* The Genetic Study of Atherosclerosis Risk (GeneSTAR) is an ongoing, prospective family-based study designed to explore environmental, phenotypic, and genetic causes of premature cardiovascular disease. Participants were recruited from European- and African-American families (*n* = 891) identified from probands who were hospitalized for a coronary disease event prior to 60 years of age in any of 10 Baltimore, Maryland area hospitals. Apparently healthy siblings of the probands, offspring of the siblings and probands, and the co-parents of the off-spring were screened for traditional coronary disease and stroke risk factors as part of a study of platelet function prior to and following a 2-week trial of 81 mg/day of aspirin from 2003 to 2006[3,34]. All measures described here were obtained prior to the commencement of aspirin. Exclusion criteria included: 1) any coronary heart disease or vascular thrombotic event, 2) any bleeding disorder or hemorrhagic event (e.g., stroke or gastrointestinal bleed), 3) current use of any anticoagulants or antiplatelet agents (i.e., warfarin, persantin, clopidogrel), 4) current use of chronic

**Fig. 3 Protein-coding variant effects in SVEP1 on ADP platelet activation, protein domains and cross-species conservation of 229G, 2702D and surrounding sequence.** Association of aggregated rare deleterious coding variants in *SVEP1* and ADP-induced platelet aggregation. *P* values in panel **A** are from two-sided score tests with no adjustment for multiple testing. **A** Using a *leave-one-out* approach, we identified a rare coding variant (rs61751937) that explains most of the association. The first orange dot represents the overall gene-based test including the full set of 64 variants. Subsequent orange dots represent the -log10(P) of the gene-based test when the specific labeled variant was left out, and blue bars represent minor allele frequency of specific variant being left out. **B** Schematic protein structure of SVEP1. rs61751937 substitutes glycine for arginine at position 229. Another variant in *SVEP1* has been associated with coronary artery disease which substitutes glycine for aspartic acid at position 2702. **C** Using UniProt, a total of 98 orthologs were identified for the largest human SVEP1 protein isoform and aligned. Alignments were visualized in MAFFT (v.7, https://mafft.cbrc.jp/alignment/server/, Katoh et al. 2017)[59] with ClustalW coloring. Both amino acids 229G and 2702D are highly conserved across diverse species, as well as their surrounding protein domains. The sequence identifiers and genus and species are given in Supplementary Data 3.

or acute nonsteroidal anti-inflammatory agents, including COX-2 inhibitors that could not be discontinued, 5) recent active gastrointestinal disorder, 6) current pharmacotherapy for a gastrointestinal disorder, 7) pregnancy or risk of pregnancy during the trial, 8) recent menorrhagia, 9) known aspirin intolerance or allergic side effects, 10) serious medical disorders, (e.g., autoimmune diseases, renal or hepatic failure, cancer or HIV-AIDS), 11) current chronic or acute use of gluco-corticosteroid therapy or any drug that may interfere with the measured outcomes, 12) serious psychiatric disorders, and, 13) inability to independently make a decision to participate. Of the 3003 participants in the aspirin trial, 1786 were selected for whole-genome sequencing (WGS) in the Trans-Omics for Precision Medicine (TOPMed) Program based on 1) complete platelet function phenotyping and 2) largest family size.

*Framingham Heart Study.* The Framingham Heart Study (FHS) is a longitudinal family-based study that started to recruit participants of European ancestry in 1948 and now is on its third generation of participants. The Original cohort (first generation) contains 5209 participants, the Offspring cohort (second generation), began to recruit in 1971, contains 5124 participants, and the Third Generation cohort, began to recruit in 2002, contains 4095 participants. In the present study, we use data from the Offspring cohort[10]. For FHS, aspirin use was determined based on arachidonic acid and review of platelet aggregation curves.

*Old Order Amish (OOA).* As part of the Amish Complex Disease Research Program, a prospective cohort trial examining the relationship between genetic variants and agonist-induced platelet function at baseline and in response to clopidogrel and aspirin was performed. Characteristics of this cohort have been described previously[35]. Briefly, Amish participants who were over age 20, generally healthy, and agreed to discontinue the use of medications, supplements, and vitamins for at least one week prior to study initiation were eligible for recruitment. Medical and family histories, anthropometry, physical examinations, and blood samples were obtained after an overnight fast. All measures described here were obtained prior to clopidogrel or aspirin administration. Participants were excluded from participation if any of the following criteria were met: 1) currently pregnant or breastfeeding, 2) history of a bleeding disorder or major spontaneous bleed, 3) severe hypertension (bp >160/95 mm Hg), 4) coexisting malignancy, 5) creatinine >2.0 mg/dl, 6) AST or ALT >2 times the upper limit of normal, 7) Hct <32%, 8) TSH <0.4 or >5.5 mIU/L, 9) platelet count >500,000/ul or <75,000/ul, 10) surgery within the last 6 months, 11) allergy to aspirin or clopidogrel, or 12) unwilling or unable to discontinue any medications that may interfere with the results of the study outcomes.

Written informed consent was obtained from all participants, and each study was approved by their local review board (GeneSTAR- Johns Hopkins Institutional Review Board; FHS- Boston University Institutional Review Board; and OOA- University of Maryland, Baltimore Institutional Review Board).

**Platelet function tests and phenotype harmonization.** Methods to assess ex vivo platelet function have been described in detail previously[8,10]. In brief, blood samples were obtained after an overnight fast into 3.2% (or 3.8% in FHS) citrated vacutainer tubes. Platelet-rich and platelet-poor plasma (PRP and PPP, respectively) were isolated by centrifugation (PRP, $180 \times g$ for 15 min in GeneSTAR and OOA, $160 \times g$ for 5 min in FHS; PPP $2000 \times g$ for 10 min in GeneSTAR and OOA, $2500 \times g$ for 20 min for FHS). Light transmittance aggregometry was performed in PRP using a PAP-4 (GeneSTAR and FHS) or a PAP-8E (OOA) aggregometer after stimulation with ADP, epinephrine, or collagen using PPP as a referent. In GeneSTAR, maximal aggregation (% aggregation) was recorded for periods of 5 min after stimulation with ADP (2.0 and 10.0 μM, Chronolog Corp, Haverton, PA) or epinephrine (2.0 and 10.0 μM, Chronolog Corp, Haverton, PA); and lag time to initiation of aggregation was recorded after stimulation with equine tendon–derived type I collagen (1, 2, 5 and 10 μg/ml; Chronolog Corp, Haverton, PA). The same methods, agonists, and agonist concentrations were used in the OOA cohort with the exception that only one concentration of epinephrine (10 μM) was used and an extra concentration of ADP (5 μM) was tested. FHS tested aggregation for periods of 4 min after administration of ADP (1.0, 3.0, 5.0, and 10.0 μM) and 5 min after administration of epinephrine (0.5, 1.0, 3.0, 5.0 and

10.0 μM); and, lag time to aggregation was assessed after stimulation with 190 μg/ml calf skin–derived type I collagen (Bio/Data Corporation, Horsham, PA). Threshold concentrations to ADP and epinephrine ($EC_{50}$) were determined as the minimal concentration of agonist required to produce >50% aggregation.

Using an adapted two stage procedure[36], platelet aggregation traits were adjusted for age, sex and aspirin-use using linear model, and the residuals from the linear model were inverse normal transformed within each cohort. Given the difference in agonist concentrations used between GeneSTAR/OOA and FHS cohorts, predefined phenotypes were identified and harmonized across studies. For ADP, epinephrine, and collagen independently, identical or closely matching agonist concentrations, the transformed residuals were combined across studies for analysis to test for association between genetic variants and low as well as high concentrations of each agonists. In total, 19 traits were defined: three low-dose ADP traits, four high-dose ADP traits, five low-dose epinephrine traits, three high-dose epinephrine traits, two low-dose collagen traits, and two high-dose collagen traits. Additional details regarding platelet phenotype harmonization are shown in Supplementary Table 1.

*TOPMed whole-genome sequencing.* WGS was performed to an average depth of 38X using DNA isolated from blood, PCR-free library construction, and Illumina HiSeq X technology. All samples used in this set of TOPMed genomes were from Freeze 5b. Details for variant calling and quality control are described in a companion paper by Taliun et al.[37]. Briefly, variant discovery and genotype calling was performed jointly, across all the available TOPMed Freeze 5b studies, using the GotCloud pipeline resulting in a single, multi-study, genotype call set. Sample-level quality control was performed to check for pedigree errors, discrepancies between self-reported and genetic sex, and concordance with prior genotyping array data.

*Variant annotation.* Variant annotation was performed using the WGSA7[38] and dbNSFP[39]. Variants were annotated as exonic, splicing, ncRNA, UTR5, UTR3, intronic, upstream, downstream, or intergenic. Exonic variants were further annotated as frameshift insertion, frameshift deletion, frameshift block substitution, stopgain, stoploss, nonframeshift insertion, nonframeshift deletion, non-frameshift block substitution, nonsynonymous variant, synonymous variant, or unknown. Additional scores available included REVEL[40], MCAP[41] or CADD[42] effect prediction algorithms.

**Single variant tests for association.** All analyses in this study were performed on the Analysis Commons[43]. Variants with minor allele count (MAC) of at least 5 and depth of coverage (DP) of at least 10 were selected for single variant analyses. The GWAS were conducted using GENetic EStimation and Inference in Structured samples (GENESIS)[44,45] apps on Analysis Commons. GENESIS uses a linear mixed model with a genetic relationship matrix (GRM) that is robust to population structure and can account for known or cryptic relatedness. The combined transformed residuals were used to conduct null model analysis adjusting for cohort indicators using genesis_nullmodel app (https://github.com/AnalysisCommons/). Single variant analysis and Sequence Kernel Association Test (SKAT) gene-based analyses were performed using *genesis_tests* app.

We used $p < 5 \times 10^{-8}$ as our genome-wide significant threshold in single variant analysis including conditional analysis for identifying independent signals. Conditional analysis was conducted by selecting the genome-wide significant variant with lowest $p$ value on a chromosome for conditioning and performing single variant analysis on the same chromosome. The procedure was repeated until no genome-wide significant variant is identified in conditional analysis by chromosome. Any variant surpassing genome-wide significance in conditional analysis was considered to be an additional signal independent of conditioned variant(s).

*Gene-based coding variant tests for association.* To improve the power to identify rare variants in coding regions, we aggregated deleterious rare coding variants in 17,774 protein-coding genes and then tested for association with platelet aggregation phenotypes. To enrich for functional variants, only variants with a "deleterious" consequence for its corresponding gene or genes (http://www.ensembl.org/info/genome/variation/predicted_data.html#consequences), were included. For each

protein-coding gene, a set of rare coding variants (MAF < 0.05) was constructed, which was composed of all stop-gain, stop-loss and frameshift variants as well as exonic missense variants that fulfilled one of these criteria: 1) REVEL score >0.5, 2) M_CAP score was "Deleterious", or 3) CADD score >30. The protein coding variant groupings were tested using SKAT with the beta-distribution parameters of 1 and 25 as proposed by Wu et al.[19]. Significance was evaluated for each platelet aggregation trait after Bonferroni correction $(0.05/17,744 = 2.82 \times 10^{-6})$.

Next we sought to determine which rare deleterious variants in each significant gene were driving the association signal. We iterated through the variants and removed one variant at a time (*leave-one-out* approach) and repeated the SKAT analysis. If a variant made a large contribution to the original association signal, one would expect the signal to significantly weaken with removal of the variant from the gene set.

*Super-enhancer based rare variant tests for association.* We investigated rare non-coding variants with putative regulatory potential by focusing on megakaryocyte-specific super enhancers (MK SEs). The published MK SEs[20] were called based on regions identified as enhancers through genome segmentation across a set of six histone modifications (H3K4me1, H3K4me3, H3K9me3, H3K27ac, H3K27me3 and H3K36me3) profiled in the BLUEPRINT project[46], aggregating together elements within 12.5 kb and then ranking upon H3K27ac signal with the ROSE algorithm[47,48]. We annotated rare (MAF < 0.05) non-coding variants located within megakaryocyte DNase I Hypersensitivity Site (DHS) peaks generated by BLUEPRINT and subset to those overlapping with MK SEs. We then applied SKAT, aggregating these non-coding variants on the set of MK SEs (n = 1065) to identify the association of these regulatory elements with platelet aggregation phenotypes. Significance was evaluated for each platelet aggregation trait after Bonferroni correction $(0.05/1065 = 4.69 \times 10^{-5})$. When a gene was identified, we conducted leave-one-out analysis to identify if a variant(s) contributed to the observed signal.

*Replication.* Additional samples from each cohort which were not included in TOPMed and therefore not included in the discovery analyses were used to replicate the genome-wide significant variants identified in the discovery analyses. In brief, genotype imputation and replication analyses were conducted by each cohort, and then meta-analysis was used to combine cohort replication analysis results. For signals identified in our gene-based tests, instead of conducting gene-based replication analysis, we replicated the single rare variants that drove the signals and that were identified from leave-one-out analyses, as not all selected rare variants had good imputation quality and were available in each cohort[49]. Each cohort independently, and separately by race for GeneSTAR, imputed the 22 autosomes using the TOPMed Freeze5b reference panel with Minimac4[50]. We implemented sample quality control procedures (excluding duplicate/reference samples and gender mismatches) and genotyping quality control procedures (excluding variants with call rate <95%, HWE $p$ value $<10^{-6}$, or MAF < 0.5%). After lifting over to build 38, non-ambiguous strand flips were resolved and ambiguous strand flips were removed. Post-imputation quality control was performed considering 6 MAF bins and using an imputation $R^2$ cutoff between 0.3 and 0.8, incrementing by 0.1 such that the mean $R^2$ exceeded 0.8 for each MAF bin. A maximum of 2009 samples were available for replication analyses: 395 OOA, 1289 FHS, 246 GS-EA, and 100 GS-AA.

GS-EA, GS-AA and OOA used the same statistical methods as the discovery analysis for the replication analysis, again using the Analysis Commons. In brief, the platelet traits were adjusted for age, sex and aspirin-use using a linear model separately in each cohort (OOA, GS-EA, GS-AA), and the residuals from these models were then inverse normal transformed within each cohort. Using GENESIS, null models were fitted within each cohort using a linear mixed model with a GRM and no covariates. Using these null models, single variant analyses were performed using GENESIS, again within each cohort. FHS used a linear mixed effects model with a relationship coefficient matrix that accounts for familial correlation implemented in the *coxme*[51] R package to conduct replication analysis, where linear model adjusting for age, sex and aspirin use was used to obtain residuals, the residuals were inverse normalized and then used for testing genetic association. Replication meta-analyses were performed using the sample size weighted approach implemented in METAL[52]. An imputation quality filter of $R^2 \geq 0.7$ was applied prior to meta-analysis, that is, at a particular variant, any cohorts with $R^2 < 0.7$ did not contribute to the variant's meta-analysis. Replication p-values were reported based on a one-sided test since the same effect direction was the expected result to reject the null hypothesis.

*Extension of results to an independent GWAS cohort.* The Caerphilly Prospective Study (CaPS) participants were relatively healthy, middle aged males at recruitment and their ages at time of blood draw (Exam 2) ranged from 47 to 66. The extent of platelet aggregation to three agonists was measured in PRP adjusted to 300,000 platelets/µl with autologous PPP[53]. Agonists included collagen (42.7 µg/ml), ADP (0.725 µM/l), and full-length thrombin (0.056 units/ml). The maximal optical density increase due to platelet aggregation was measured and expressed as a proportion of the difference in optimal density between PRP and PPP. Genotyping was performed with the Affymetrix UK BioBank array using the Affymetrix Axiom Analysis program. Following sample and genotyping quality control, imputation

on 22 autosomes was performed using the HRC 1.1 reference panel, resulting in ~7.6 million variants with MAF > 0.01 and $R^2 > 0.4$. GWAS was performed with the Efficient Mixed-Model Association eXpedited (EMMAX) package. For each trait, a linear model was constructed adjusting for age and medication use (anticoagulant, antiplatelet, antilipid, hypoglycemics) and single variant analyses were performed on transformed platelet reactivity values. Maximum sample sizes and phenotype transformations were as follows: ADP (n = 1177, natural log), thrombin (n = 1183, square root), and collagen (n = 811, cube root). Although the agonists differed in some cases in dose or type from the discovery efforts, we had the prior hypothesis that platelet-reactivity increasing alleles for one agonist are likely to be reactivity increasing for other agonists/doses. Note that in CaPS collagen maximal aggregation was measured, and collagen lag time unavailable, thus, the expected effect direction would be opposite to our discovery analyses (as observed for PEAR1 rs12041331 in Supplementary Table 4 versus collagen lag time discovery results in Supplementary Data 1). Association extension results in CaPS are reported with beta, standard-error, and one-sided $p$ values relative to the hypothesized direction.

*Co-localization of expression quantitative trait loci (eQTL) signatures from platelets in GeneSTAR European Americans.* A subset of 180 TOPMed GeneSTAR European Americans samples also had RNA-seq data generated using platelets. eQTL analysis was performed as previously described[54]. Here, formal Bayesian co-localization was performed using the *coloc*[55–57] package in R for each of the 16 independent loci (Table 1) against all gene transcripts where there was at least one SNP with an eQTL $p$ value $p < 0.003125$ (0.05/16) for the specific gene within 20 KB of the peak variant. This yielded 10 locus-gene pairs (Supplementary Table 4). *coloc* tests five mutually exclusive hypotheses: H0, no GWAS and no eQTL association; H1, association with GWAS, but no eQTL; H2, association with eQTL, but no GWAS; H3, eQTL and GWAS association, but with two independent causal variants; and H4, shared causal variants for both eQTL and GWAS. The main interest is to assess whether there is a shared causal variant between eQTL and GWAS (i.e., H4). The package provides five posterior probabilities for these hypotheses (PP0, PP1, PP2, PP3, and PP4) and PP4 of >75% is considered evidence of a colocalization of GWAS and eQTL. Posterior probabilities for individual variants were evaluated once PP4 was met.

**Allele-specific and transcription-factor enriched enhancer assays**

*Cell culture.* K562 is a lymphoblastoid human erythroleukemia cell line derived from a female donor. It is a suspension cell line. K562 cells were cultured and maintained in RPMI 1640 media supplemented with 10% FBS (Sigma-Aldrich), Pen/Strep and L-Gln. Cultures were maintained in a humidified environment at 37 °C with 5% $CO_2$. K562 cells were passaged every 24–48 h. HEK293 cells were cultured and maintained at low passage in DMEM media supplemented with 5% FBS (Sigma-Aldrich). Cultures were maintained in a humidified environment at 37 °C with 5% $CO_2$.

*Lentivirus production.* For Lentivirus production following vectors were used: pInducer-21 lentiviral vector (Addgene), pMD2.G envelope plasmid (Addgene), psPAX2 packaging plasmid (Addgene). 293T-17 cells (ATCC) were cultured and maintained at low passage in DMEM media supplemented with 5% FBS (Sigma-Aldrich). Cultures were maintained in a humidified environment at 37 °C with 5% $CO_2$. 293T-17 cells were passaged every 24–48 h. Lentiviral plasmids possessing open reading frames of GFP (Empty), POLR2A, NRF1, CTCF, FOSL1, GATA1, GATA2, CEBPB, and NFE2 were cloned into pInducer-21 lentiviral vector. For lentivirus production, 293T-17 cells (ATCC) were transfected with third generation packaging plasmids pMD2.G and psPAX2 (Addgene) and lentiviral plasmids POLR2A, NRF1, CTCF, FOSL1, GATA1, GATA2, CEBPB, and NFE2. Viruses were harvested 48 h post transfections and concentrated by ultracentrifugation at $71,286 \times g$ for 2 h at 4 °C. Viruses were titrated by serial dilution on 293 T cells using GFP as an indicator.

*RNA extraction, reverse transcription, and RT-qPCR.* RNA extraction from variously transduced HEK293 and K562 cells was performed using an RNAeasy kit (Qiagen). Reverse transcription was performed using Superscript III (Invitrogen), using Oligo (dT) 15 primer. Quantitative PCRs were performed in triplicate with Taqman primer prober assays, shown in Methods Table 1 and CFX96 real-time PCR detection system (Bio-rad). Target transcript abundance was calculated relative to ACTB (reference gene) using the 2-ΔΔCT method. Gene specific primer pairs are present in methods section Oligonucleotides.

*Enhancer function reporter assay.* The following vectors were used in this protocol: pGL3 luciferase reporter (Promega), pGL4.74[hRluc/TK] control vector (Promega). ~200–300 base pair non-coding regions of RGS18, ADRA2A and PEAR1 and the associated alleles surrounding the various SNP variants were cloned into the pGL3 luciferase vector. We created two modified constructs to assess functionality of the various loci: wild-type loci carrying no SNP and knock-in of the various SNPs into the respective loci. The constructs were generated via Vectorbuilder. Constructs were sequenced to confirm the expected genotype and to ensure no off-target mutations were introduced. Dual luciferase reporter assays were performed as

described previously with minor modifications[11,58]. Briefly, GFP, POLR2A, NRF1, CTCF, FOSL1, GATA1, GATA2, CEBPB, or NFE2 overexpressing HEK293 cells or K562 cells were co-transfected with one of the two pGL3 luciferase vectors described above as well pGL3 control according to the manufacturer's instructions. Firefly and Renilla luciferase reporter activity of cell extracts were measured using the Dual-Glo Luciferase Assay System (Promega) on a microplate reader according to the manufacturer's instructions. Each treatment was performed in duplicate and the experiment was repeated three times. Assay primer information is given in Supplementary Table 12.

**Phenome-wide association study (PheWAS)**. The 16 genome-wide significant variants identified from WGS in our discovery cohort (Table 1) as well as significant variants identified in gene-based tests, were examined against clinical phenotypes in the UKBB and BioVU cohorts where available. We queried UKBB GWAS results using SAIGE calculated summary statistics (http://www.nealelab.is/uk-biobank/faq). The BioVU repository contains >250,000 DNA samples obtained from discarded blood samples of consented patients at Vanderbilt University Medical Center (Nashville, TN). De-identified DNA samples in the BioVU repository are linked to 1543 clinical diagnostic codes. Of these 1543 clinical diagnostic codes, we identified 71 diagnoses for which platelet function could be in the pathophysiologic pathway to disease expression. Numerous overlapping disease processes were represented among these 71 codes, which we further categorized into the following 6 phenotypes: arterial thrombosis (30 codes), venous thrombosis (3 codes), hypercoagulable state (2 codes), platelet (5 codes), bleeding (26 codes), and anti-thrombotic medication usage (5 codes). The phecodes were matched between UKBB and BioVU for corresponding allelic results.

**Reporting summary**. Further information on research design is available in the Nature Research Reporting Summary linked to this article.

## Data availability

TOPMed WGS variant calls are available for all samples through the Database of Genotypes and Phenotypes (dbGaP) under accession number phs001218 for GeneSTAR, phs000956 and phs000391 for OOA and phs000974 for Framingham. Phenotype data for GeneSTAR, OOA and Framingham are also available through this mechanism. Summary statistics are being deposited in the TOPMed GSR (Genomic Summary Results) site. eQTL analysis results used in the co-localization analysis are hosted on a website at: http://www.biostat.jhsph.edu/~kkammers/GeneSTAR/.

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

## Acknowledgements

Whole genome sequencing (WGS) for the Trans-Omics in Precision Medicine (TOPMed) program was supported by the National Heart, Lung and Blood Institute (NHLBI). WGS for GeneSTAR (Genetic Study of Atherosclerosis Risk) was performed at Macrogen, Illumina, and the Broad Institute of MIT and Harvard (HHSN268201500014C). WGS for the Old Order Amish (Genetics of Cardiometabolic Health in the Amish) was performed at the Broad Institute of MIT and Harvard (3R01HL121007-01S1). WGS for The Framingham Heart Study (Whole Genome Sequencing and Related Phenotypes in the Framingham Heart Study) was performed at the Broad Institute of MIT and Harvard (HHSN268201500014C). Centralized read mapping and genotype calling, along with variant quality metrics and filtering were provided by the TOPMed Informatics Research Center (3R01HL-117626-02S1). Phenotype harmonization, data management, sample-identity QC, and general study coordination, were provided by the TOPMed Data Coordinating Center (3R01HL-120393-02S1). For the Old Order Amish this investigation was supported by National Institutes of Health grants U01 GM074518, U01 HL105198, R01 HL137922, R01 HL121007, and the University of Maryland Mid-Atlantic Nutrition and Obesity Research Center (P30 DK072488). GeneSTAR was supported by the National Institutes of Health/National Heart, Lung, and Blood Institute (U01 HL72518, HL087698, HL112064, HL11006, HL118356) and by a grant from the National Institutes of Health/National Center for Research Resources (M01-RR000052) to the Johns Hopkins General Clinical Research Center. The Framingham Heart Study is conducted and supported by the NHLBI in collaboration with Boston University (Contract No. N01-HC-25195), its contract with Affymetrix, Inc., for genome-wide genotyping services (Contract No. N02-HL-6-4278 and Contract No. HHSN268201500001I). MHC, BATC and ADJ were supported by NHLBI Intramural funding. The Caerphilly Prospective study was undertaken by the former MRC Epidemiology Unit (South Wales) and was funded by the Medical Research Council of the UK. The data archive is maintained by the School of Social and Community Medicine, University of Bristol. This study makes use of data generated by the BLUEPRINT Consortium. A full list of the investigators who contributed to the generation of the data is available from www.blueprint-epigenome.eu. Funding for the project was provided by the European Union's Seventh Framework Programme (FP7/2007-2013) under grant agreement no 282510 BLUEPRINT. Additional support came from the National Blood Foundation/American Association of Blood Banks (FP01021164), the National Institute of Diabetes and Digestive and Kidney Diseases (NIDDK; U54DK110805) and the National Research Service Award (NRSA)'s Joint Program in Transfusion Medicine (T32 4T32HL066987-15 to A.B.). BioVU resource analyses were supported by National Institutes of Health/National Genome Research Institute grant U01HG009086. The views expressed in this manuscript are those of the authors and do not necessarily represent the views of the National Heart, Lung, and Blood Institute; the National Institutes of Health; or the U.S. Department of Health and Human Services.

## Author contributions

A.R.K., M.-H.C., B.A.T.R. led equal roles in analysis, interpretation and writing of the paper. J.P.L., R.A.M., A.D.J. led equal senior roles for the study. A.R.K. wrote the first draft of manuscript with contribution and editing from M.-H.C., B.A.T.R., L.R.Y., M.A.T., J.A.B., L.C.B., N.F., J.P.L., R.A.M. and A.D.J. Genome wide analyses were performed by A.R.K., B.A.T.R., B.J.G., L.R.Y., M.-H.C. and K.R. RNA-sequencing and eQTL analyses were performed by R.A.M., Kai.K., M.A.T., I.R., L.R.Y., A.R.K., Kan.K. and K.I. Imputation of genomic data and replication analyses were performed by M.-H.C., B.A.T.R., L.R.Y., B.J.G., A.P., L.A.C., and M.H.K. L.R.Y., N.F., L.C.B. and R.A.M. were involved in the guidance, collection and analysis for Genetic Study of Atherosclerosis Risk (GeneSTAR) phenotype data. B.J.G., K.R., B.D.M., J.P.L., J.R.O. and A.R.S. were involved in the guidance, collection and analysis for Older Order Amish Study (OAA) phenotype data. B.A.T.R., M.-H.C. and A.D.J. were involved in the guidance and analysis for Framingham Heart Study (FHS) phenotype data. T.M.S. and A.B. established the imMKCL system, and A.B. designed all functional experiments with input from B.A.T.R. and A.D.J. and carried them out. X.Z., Q.W. and B.L. carried out BioVU pheWAS analyses. A.D.J. funded genotyping of the CaPS cohort. M.-H.C. performed genotype QC, calling and imputation of the CaPS cohort with input from A.D.J., B.A.T.R., M.-H.C. and A.D.J. conducted CaPS genotype-phenotype analyses.

## Funding

## Competing interests

The authors declare no competing interests.

## Additional information

## NHLBI Trans-Omics for Precision (TOPMed) Consortium

Namiko Abe[19], Goncalo Abecasis[20], Francois Aguet[21], Christine Albert[22], Laura Almasy[23], Alvaro Alonso[24], Seth Ament[25], Peter Anderson[26], Pramod Anugu[27], Deborah Applebaum-Bowden[28], Kristin Ardlie[21], Dan Arking[29], Donna K. Arnett[30], Allison Ashley-Koch[31], Stella Aslibekyan[32], Tim Assimes[33], Paul Auer[34], Dimitrios Avramopoulos[29], Najib Ayas[35], Adithya Balasubramanian[36], John Barnard[37], Kathleen Barnes[38], R. Graham Barr[39], Emily Barron-Casella[29], Lucas Barwick[40], Terri Beaty[29], Gerald Beck[37], Diane Becker[29], Lewis Becker[29], Rebecca Beer[41], Amber Beitelshees[25], Emelia Benjamin[42], Takis Benos[43], Marcos Bezerra[44], Larry Bielak[20], Joshua Bis[26], Thomas Blackwell[20], John Blangero[45], Eric Boerwinkle[46], Donald W. Bowden[47], Russell Bowler[48], Jennifer Brody[26], Ulrich Broeckel[49], Jai Broome[26], Deborah Brown[46], Karen Bunting[19], Esteban Burchard[50], Carlos Bustamante[33], Erin Buth[26], Brian Cade[51], Jonathan Cardwell[52], Vincent Carey[51], Julie Carrier[53], Cara Carty[54], Richard Casaburi[55], Juan P. Casas Romero[51], James Casella[29], Peter Castaldi[51], Mark Chaffin[21], Christy Chang[25], Yi-Cheng Chang[56], Daniel Chasman[51], Sameer Chavan[52], Bo-Juen Chen[19], Wei-Min Chen[57], Yii-DerIda Chen[58], Michael Cho[51], Seung Hoan Choi[21], Lee-Ming Chuang[56], Mina Chung[37], Ren-Hua Chung[59], Clary Clish[21], Suzy Comhair[37], Matthew Conomos[26], Elaine Cornell[60], Adolfo Correa[27], Carolyn Crandall[55], James Crapo[48], L. Adrienne Cupples[61], Joanne Curran[45], Jeffrey Curtis[20], Brian Custer[62], Coleen Damcott[25], Dawood Darbar[63], Sean David[64], Colleen Davis[26], Michelle Daya[52], Mariza de Andrade[65], Lisa de las Fuentes[66], Paul de Vries[46], Michael DeBaun[67], Ranjan Deka[68], Dawn DeMeo[51], Scott Devine[25], Huyen Dinh[36], Harsha Doddapaneni[36], Qing Duan[69], Shannon Dugan-Perez[36], Ravi Duggirala[70], Jon Peter Durda[60], Susan K. Dutcher[66], Charles Eaton[71], Lynette Ekunwe[27], Adel El Boueiz[72], Patrick Ellinor[73], Leslie Emery[26], Serpil Erzurum[37], Charles Farber[57], Jesse Farek[36], Tasha Fingerlin[48], Matthew Flickinger[20], Myriam Fornage[46], Nora Franceschini[69], Chris Frazar[26], Mao Fu[25], Stephanie M. Fullerton[26], Lucinda Fulton[66], Stacey Gabriel[21], Weiniu Gan[41], Shanshan Gao[52], Yan Gao[27], Margery Gass[74], Heather Geiger[19], Bruce Gelb[75], Mark Geraci[43], Soren Germer[19], Robert Gerszten[76], Auyon Ghosh[51], Richard Gibbs[36], Chris Gignoux[33], Mark Gladwin[43], David Glahn[77], Stephanie Gogarten[26], Da-Wei Gong[25], Harald Goring[78], Sharon Graw[38], Kathryn J. Gray[79], Daniel Grine[52], Colin Gross[20], C.Charles Gu[66], Yue Guan[25], Xiuqing Guo[58], Namrata Gupta[21], David M. Haas[80], Jeff Haessler[75], Michael Hall[27], Yi Han[36], Patrick Hanly[81], Daniel Harris[82], Nicola L. Hawley[83], Jiang He[84], Ben Heavner[26], Susan Heckbert[26], Ryan Hernandez[50], David Herrington[47], Craig Hersh[51], Bertha Hidalgo[32], James Hixson[46], Brian Hobbs[51], John Hokanson[52], Elliott Hong[25], Karin Hoth[85], Chao Agnes Hsiung[59], Jianhong Hu[36], Yi-Jen Hung[86], Haley Huston[87], Chii Min Hwu[88], Marguerite Ryan Irvin[32], Rebecca Jackson[89], Deepti Jain[26], Cashell Jaquish[41], Jill Johnsen[87], Andrew Johnson[41], Craig Johnson[26], Rich Johnston[24], Kimberly Jones[29], Hyun Min Kang[20], Robert Kaplan[90], Sharon Kardia[20], Shannon Kelly[50], Eimear Kenny[75], Michael Kessler[25], Alyna Khan[26], Ziad Khan[36], Wonji Kim[72], John Kimoff[91], Greg Kinney[93], Barbara Konkle[87], Charles Kooperberg[75], Holly Kramer[94], Christoph Lange[92], Ethan Lange[52], Leslie Lange[52], Cathy Laurie[26], Cecelia Laurie[26], Meryl LeBoff[51], Jiwon Lee[51], Sandra Lee[36], Wen-Jane Lee[88], Jonathon LeFaive[20], David Levine[26], Dan Levy[41], Joshua Lewis[25], Xiaohui Li[88], Yun Li[69], Henry Lin[58], Honghuang Lin[61], Xihong Lin[92], Simin Liu[71], Yongmei Liu[31], Yu Liu[33], Ruth J. F. Loos[75], Steven Lubitz[72], Kathryn Lunetta[61], James Luo[41], Ulysses Magalang[95], Michael Mahaney[45], Barry Make[29], Ani Manichaikul[58], Alisa Manning[96], JoAnn Manson[51], Lisa Martin[97], Melissa Marton[19], Susan Mathai[52], Rasika Mathias[29], Susanne May[26], Patrick McArdle[25], Merry-Lynn McDonald[32], Sean McFarland[72], Stephen McGarvey[71], Daniel McGoldrick[26], Caitlin McHugh[26], Becky McNeil[98], Hao Mei[27], James Meigs[72], Vipin Menon[36], Luisa Mestroni[38], Ginger Metcalf[36], Deborah A. Meyers[99], Emmanuel Mignot[100], Julie Mikulla[41], Nancy Min[27], Mollie Minear[101], Ryan L. Minster[43], Braxton D. Mitchell[25], Matt Moll[51], Zeineen Momin[36], May E. Montasser[25], Courtney Montgomery[102], Donna Muzny[36], Josyf C. Mychaleckyj[57], Girish Nadkarni[76], Rakhi Naik[29],

Take Naseri[103], Pradeep Natarajan[21], Sergei Nekhai[104], Sarah C. Nelson[26], Bonnie Neltner[52], Caitlin Nessner[36], Deborah Nickerson[26], Osuji Nkechinyere[36], Kari North[69], Jeff O'Connell[105], Tim O'Connor[25], Heather Ochs-Balcom[106], Geoffrey Okwuonu[36], Allan Pack[107], David T. Paik[33], Nicholette Palmer[47], James Pankow[108], George Papanicolaou[41], Cora Parker[109], Gina Peloso[61], Juan Manuel Peralta[69], Marco Perez[33], James Perry[25], Ulrike Peters[73], Patricia Peyser[20], Lawrence S. Phillips[24], Jacob Pleiness[20], Toni Pollin[25], Wendy Post[29], Julia Powers Becker[52], Meher Preethi Boorgula[52], Michael Preuss[75], Bruce Psaty[26], Pankaj Qasba[41], Dandi Qiao[51], Zhaohui Qin[24], Nicholas Rafaels[52], Laura Raffield[69], Mahitha Rajendran[36], Vasan S. Ramachandran[62], D. C. Rao[66], Laura Rasmussen-Torvik[110], Aakrosh Ratan[58], Susan Redline[51], Robert Reed[25], Catherine Reeves[19], Elizabeth Regan[48], Alex Reiner[111], Muagututi'a Sefuiva Reupena[112], Ken Rice[26], Stephen Rich[58], Rebecca Robillard[113], Nicolas Robine[19], Dan Roden[67], Carolina Roselli[21], Jerome Rotter[58], Ingo Ruczinski[29], Alexi Runnels[19], Pamela Russell[52], Sarah Ruuska[87], Kathleen Ryan[25], Ester Cerdeira Sabino[114], Danish Saleheen[39], Shabnam Salimi[25], Sejal Salvi[36], Steven Salzberg[29], Kevin Sandow[58], Vijay G. Sankaran[72], Jireh Santibanez[36], Karen Schwander[66], David Schwartz[52], Frank Sciurba[43], Christine Seidman[115], Jonathan Seidman[115], Frederic Series[116], Vivien Sheehan[24], Stephanie L. Sherman[24], Amol Shetty[25], Aniket Shetty[52], Wayne Hui-Heng Sheu[88], M. Benjamin Shoemaker[67], Brian Silver[117], Edwin Silverman[51], Robert Skomro[118], Albert Vernon Smith[20], Jennifer Smith[20], Josh Smith[26], Nicholas Smith[26], Tanja Smith[19], Sylvia Smoller[90], Beverly Snively[47], Michael Snyder[33], Tamar Sofer[51], Nona Sotoodehnia[26], Adrienne M. Stilp[26], Garrett Storm[52], Elizabeth Streeten[25], Jessica Lasky Su[51], Yun Ju Sung[67], Jody Sylvia[51], Adam Szpiro[26], Daniel Taliun[20], Hua Tang[33], Margaret Taub[29], Kent D. Taylor[58], Matthew Taylor[38], Simeon Taylor[25], Marilyn Telen[31], Timothy A. Thornton[26], Machiko Threlkeld[26], Lesley Tinker[73], David Tirschwell[26], Sarah Tishkoff[107], Hemant Tiwari[32], Catherine Tong[26], Russell Tracy[60], Michael Tsai[108], Dhananjay Vaidya[29], David Van Den Berg[119], Peter VandeHaar[20], Scott Vrieze[108], Tarik Walker[52], Robert Wallace[85], Avram Walts[52], Fei Fei Wang[26], Heming Wang[120], Jiongming Wang[20], Karol Watson[55], Jennifer Watt[36], Daniel E. Weeks[43], Bruce Weir[26], Scott T. Weiss[51], Lu-Chen Weng[72], Jennifer Wessel[80], Cristen Willer[20], Kayleen Williams[26], L. Keoki Williams[121], Carla Wilson[51], James Wilson[122], Lara Winterkorn[19], Quenna Wong[26], Joseph Wu[33], Huichun Xu[25], Lisa Yanek[29], Ivana Yang[52], Ketian Yu[20], Seyedeh Maryam Zekavat[21], Yingze Zhang[43], Snow Xueyan Zhao[48], Wei Zhao[20], Xiaofeng Zhu[123], Michael Zody[19] & Sebastian Zoellner[20]

[19]New York Genome Center, New York, NY, USA. [20]University of Michigan, Ann Arbor, MI, USA. [21]Broad Institute, Cambridge, MA, USA. [22]Cedars Sinai, Boston, MA, USA. [23]Children's Hospital of Philadelphia, University of Pennsylvania, Philadelphia, PA, USA. [24]Emory University, Atlanta, GA, USA. [25]University of Maryland, Baltimore, MD, USA. [26]University of Washington, Seattle, WA, USA. [27]University of Mississippi, Jackson, MS, USA. [28]National Institutes of Health, Bethesda, MD, USA. [29]Johns Hopkins University, Baltimore, MD, USA. [30]University of Kentucky, Lexington, KY, USA. [31]Duke University, Durham, NC, USA. [32]University of Alabama, Birmingham, LA, USA. [33]Stanford University, Stanford, CA, USA. [34]University of Wisconsin Milwaukee, Milwaukee, WI, USA. [35]Providence Health Care, Vancouver, BC, Canada. [36]Baylor College of Medicine Human Genome Sequencing Center, Houston, TX, USA. [37]Cleveland Clinic, Cleveland, OH, USA. [38]University of Colorado Anschutz Medical Campus, Aurora, OH, USA. [39]Columbia University, New York, NY, USA. [40]The Emmes Corporation, Rockville, MD, USA. [41]National Heart, Lung, and Blood Institute, National Institutes of Health, Bethesda, MD, USA. [42]Boston University, Massachusetts General Hospital, Boston, MA, USA. [43]University of Pittsburgh, Pittsburgh, PA, USA. [44]Fundação de Hematologia e Hemoterapia de Pernambuco - Hemope, Recife, BR, Brazil. [45]University of Texas Rio Grande Valley School of Medicine, Brownsville, TX, USA. [46]University of Texas Health at Houston, Houston, TX, USA. [47]Wake Forest Baptist Health, Winston-Salem, MA, USA. [48]National Jewish Health, Denver, CO, USA. [49]Medical College of Wisconsin, Milwaukee, WI, USA. [50]University of California, San Francisco, San Francisco, CA, USA. [51]Brigham & Women's Hospital, Boston, MA, USA. [52]University of Colorado at Denver, Denver, CO, USA. [53]University of Montreal, Montreal, QC, Canada. [54]Washington State University, Pullman, WA, USA. [55]University of California, Los Angeles, Los Angeles, CA, USA. [56]National Taiwan University, Taipei, TW, Taiwan. [57]University of Virginia, Charlottesville, VA, USA. [58]Lundquist Institute, Torrance, CA, USA. [59]National Health Research Institute Taiwan, Miaoli County, TW, Taiwan. [60]University of Vermont, Burlington, VT, USA. [61]Boston University, Boston, MA, USA. [62]Vitalant Research Institute, San Francisco, CA, USA. [63]University of Illinois at Chicago, Chicago, IL, USA. [64]University of Chicago, Chicago, IL, USA. [65]Mayo Clinic, Rochester, NY, USA. [66]Washington University in St Louis, St. Louis, WA, USA. [67]Vanderbilt University, Nashville, TN, USA. [68]University of Cincinnati, Cincinnati, OH, USA. [69]University of North Carolina, Chapel Hill, NC, USA. [70]University of Texas Rio Grande Valley School of Medicine, Edinburg, TX, USA. [71]Brown University, Providence, RI, USA. [72]Harvard University, Cambridge, MA, USA. [73]Massachusetts General Hospital, Boston, MA, USA. [74]Fred Hutchinson Cancer Research Center, Seattle, WA, USA. [75]Icahn School of Medicine at Mount Sinai, New York, NY, USA. [76]Beth Israel Deaconess Medical Center, Boston, MA, USA. [77]Boston Children's Hospital, Harvard Medical School, Boston, MA, USA. [78]University of Texas Rio Grande Valley School of Medicine, San Antonio,

TX, USA. [79]Mass General Brigham, Boston, MA, USA. [80]Indiana University, Indianapolis, IN, USA. [81]University of Calgary, Calgary, AB, Canada. [82]University of Maryland, Philadelphia, PA, USA. [83]Yale University, New Haven, CT, USA. [84]Tulane University, New Orleans, LA, USA. [85]University of Iowa, Iowa City, IA, USA. [86]Tri-Service General Hospital National Defense Medical Center, Taichung City, TW, Taiwan. [87]Blood Works Northwest, Seattle, WA, USA. [88]Taichung Veterans General Hospital Taiwan, Taichung City, TW, Taiwan. [89]Oklahoma State University Medical Center, Columbus, OK, USA. [90]Albert Einstein College of Medicine, New York, NY, USA. [91]McGill University, Montréal, Quebec, CA, USA. [92]Harvard School of Public Health, Boston, MA, USA. [93]University of Colorado at Denver, Aurora, CO, USA. [94]Loyola University, Maywood, IL, USA. [95]Ohio State University, Columbus, OH, USA. [96]Broad Institute, Harvard University, Massachusetts General Hospital, Cambridge, MA, USA. [97]George Washington University, Washington, WA, USA. [98]RTI International, Washington, WA, USA. [99]University of Arizona, Tucson, AZ, USA. [100]Stanford University, Palo Alto, CA, USA. [101]National Institute of Child Health and Human Development, National Institutes of Health, Bethesda, MD, USA. [102]Oklahoma Medical Research Foundation, Oklahoma City, OK, USA. [103]Ministry of Health, Government of Samoa, Apia, WS, Samoa. [104]Howard University, Washington, WA, USA. [105]University of Maryland, Balitmore, MD, USA. [106]University at Buffalo, Buffalo, NY, USA. [107]University of Pennsylvania, Philadelphia, PA, USA. [108]University of Minnesota, Minneapolis, MN, USA. [109]RTI International, Research Triangle Park, NC, USA. [110]Northwestern University, Chicago, IL, USA. [111]Fred Hutchinson Cancer Research Center, University of Washington, Seattle, WA, USA. [112]Lutia I Puava Ae Mapu I Fagalele, Apia, WS, Samoa. [113]University of Ottawa, Ottawa, QC, Canada. [114]Universidade de Sao Paulo, Sao Paulo, BR, Brazil. [115]Harvard Medical School, Boston, MA, USA. [116]Université Laval, Quebec City, CA, USA. [117]UMass Memorial Medical Center, Worcester, MA, USA. [118]University of Saskatchewan, Saskatoon, SK, Canada. [119]University of Southern California, Los Angeles, CA, USA. [120]Brigham & Women's Hospital, Mass General Brigham, Boston, MA, USA. [121]Henry Ford Health System, Detroit, MI, USA. [122]Beth Israel Deaconess Medical Center, Cambridge, MA, USA. [123]Case Western Reserve University, Cleveland, OH, USA.

