## [Peer Review File · Nature Communications]

Reviewers' Comments:

Reviewer #1:

None

Reviewer #2:

Remarks to the Author:

I have read the revised manuscript by Keremati et al. on "Genome Sequencing Unveils a New Regulatory Landscape of Platelet Reactivity" with interest.

First, I am impressed with the thoroughness of the provided rebuttal and would like to thank the authors for efforts made. Second additional analysis have been performed to address concerns which were raised by the Reviewers. The results of these new analysis addresses nearly all the issues raised in a conclusive manner. Third corrections in the text have been made based on the new results and comments made by the Reviewers. Finally experimental studies have been performed in the laboratory. These produced corroborating and new lines of evidence for some of the newly identified variants being functionally active in model cell lines. These experiments further reduce the likelihood of the reported associations between DNA variants and platelet function being false-positive ones. Furthermore this manuscript reports on the TOPMed whole genome sequencing (WGS) results in relation to platelet function. Conceptually the manuscript resets the thinking on how to reliably identify rare variants in the coding and non-coding space, which exert an effect on complex functional phenotypes of cells. The study confirms the recent observation by Van Hout et al (Nature, 2020) that ascertaining rare variants by imputation does erode the power of association studies which set out to identify rare variants with effect sizes larger than common variants. The study by Van Hout and colleagues focused on variants in the coding space because genotyping was performed by whole exome sequencing instead of whole genome sequencing. Altogether it is becoming increasingly obvious that if researchers wish to explore the genetic architecture of platelet function or other medically relevant traits in full that WGS is becoming the method of choice for genotyping. For reasons outlined above it is important that this manuscript appears in respectable peer-reviewed journal like Nature Communications. It is highly likely that future well powered association studies in larger cohorts using WGS will corroborate the observed associations which are being reported by Keremati and colleagues.

In conclusion I strongly support acceptance of the manuscript and hope that my judgement is aligned to those of the other Reviewers.

Minor Comments:

The authors may wish to reference the manuscript by Van Hout and colleagues as both studies illustrate that association studies which wish to identify the effect of rare variants must rely on high quality genotyping data produced by sequencing instead of imputation.

Figures

- Figure 1:

- o The Manhattan plots are too much size reduced and the legends cannot be deciphered
- o The Circos plot suffers from the same miniaturization problem and has therefore lost its ability to convey a clear message
- o Both figures do not respond well to 'zooming in' which I assume will be resolved when publication ready figures are produced

- Figure 2:
o The same issue applies to the left top and bottom panels of Figure 2A

- Figure 3:
o The same issue applies to Figure 3C